# Investigating Methodological Differences in the Assessment of Dendritic Morphology of Basolateral Amygdala Principal Neurons—A Comparison of Golgi–Cox and Neurobiotin Electroporation Techniques

**DOI:** 10.3390/brainsci7120165

**Published:** 2017-12-19

**Authors:** Paul M. Klenowski, Sophie E. Wright, Erica W. H. Mu, Peter G. Noakes, Nickolas A. Lavidis, Selena E. Bartlett, Mark C. Bellingham, Matthew J. Fogarty

**Affiliations:** 1Translational Research Institute, Queensland University of Technology, Brisbane 4102, Australia; paul.klenowski@umassmed.edu; 2Department of Neurobiology, University of Massachusetts Medical School, Worcester, MA 01605, USA; 3School of Biomedical Sciences, The University of Queensland, Brisbane 4072, Australia; sophiewright@live.com.au (S.E.W.); e.mu@uq.edu.au (E.W.H.M.); p.noakes@uq.edu.au (P.G.N.); lavidis@uq.edu.au (N.A.L.); 4Queensland Brain Institute, the University of Queensland, Brisbane 4072, Australia; 5Department of Physiology and Biomedical Engineering, Mayo Clinic College of Medicine, Rochester, MN 55905, USA

**Keywords:** basolateral amygdala, dendrites, Golgi–Cox, spines, neurobiotin, principal neuron

## Abstract

Quantitative assessments of neuronal subtypes in numerous brain regions show large variations in dendritic arbor size. A critical experimental factor is the method used to visualize neurons. We chose to investigate quantitative differences in basolateral amygdala (BLA) principal neuron morphology using two of the most common visualization methods: Golgi–Cox staining and neurobiotin (NB) filling. We show in 8-week-old Wistar rats that NB-filling reveals significantly larger dendritic arbors and different spine densities, compared to Golgi–Cox-stained BLA neurons. Our results demonstrate important differences and provide methodological insights into quantitative disparities of BLA principal neuron morphology reported in the literature.

## 1. Introduction

The elaborate axonal and dendritic extensions from neuronal somas provide the anatomical substrate for neuronal connections and the formation of neural networks. The neuronal dendritic field is responsible for the integration of the vast majority of synaptic inputs received by the neuron, playing a large role in determining neuronal activity [1]. Since the late 19th century, Golgi, Cajal and others have developed techniques to classify and quantitate neuronal structures en masse in the central nervous system [2,3,4,5].

The widespread application of Golgi staining methods has provided insights into the morphological characteristics of several neuronal subpopulations. More recently, the development of analytical software to evaluate neuronal properties, combined with newer methodological approaches, including intracellular dye-filling methods, genetic labeling and confocal microscopy [6,7,8,9] are now being implemented to quantitatively assess structural differences in neuronal subtypes, and to determine the genetic, environmental, behavioural and chemical effects on neuronal morphology.

The recent rise in quantitative analysis of neuronal dendritic length and complexity [9] has resulted in considerable disparities within the literature. A particularly interesting example is the difference in quantitative measurements of principal neurons in the basolateral amygdala (BLA). Using a modified rapid Golgi–Cox staining method, total dendritic arbor lengths have ranged from 1822 µm for principal neurons in 8–10-week-old Wistar rats [10], 1338 µm in 9–10-week-old Sprague-Dawley rats [11] to 2900 µm in 5-week-old Long-Evans rats [12]. We recently used neurobiotin™ (NB) filling and confocal microscopy to report the mean total dendritic arbor length as 2034 µm for NB-filled principal neurons within the BLA in 8-week-old Wistar rats [8]. Past studies using similar dye-filling methods have provided estimates for total dendrite length of 6299 to 6722 μm for principal neurons in the lateral amygdala in 17–20-day old Wistar rats [13], and 7908 μm for BLA principal cells from 8.5-week-old Sprague-Dawley rats [14]. While this wide range of arbor lengths may be due to age- or strain-related differences, it may also be due to methodological variation.

In this study, we used Golgi–Cox staining and NB-filling to provide a direct quantitative comparison of BLA principal neuron morphology. We traced BLA principal cells of 8-week-old Wistar rats, obtained using both methods, and compared dendritic arbor general morphology, branch order and Sholl [15] characteristics of BLA principal neurons. We show that the dendrites of BLA principal neurons filled with NB are significantly longer and display greater dendritic complexity, compared to Golgi–Cox-stained neurons. Additionally, we found that spine densities varied significantly in proximal and distal apical dendrites of NB-filled and Golgi–Cox-stained BLA principal neurons. Our data highlights quantitative differences in BLA principal neuron morphology, obtained with two different routinely used methods.

## 2. Methods

### 2.1. Ethics Statement

All experimental procedures were approved by The University of Queensland and the Queensland University of Technology Animal Ethics Committees and complied with State and Federal laws and guidelines.

### 2.2. Golgi–Cox Staining

Rats (*n* = 4) were euthanized by sodium pentobarbitone overdose (60–80 mg/kg, i.p., Vetcare, Brisbane, Australia), and exsanguinated intra-cardially using a heparinized needle (Sigma-Aldrich, St. Louis, MO, USA) [16]. The brain was removed and incubated in the dark for 6 days at 37 °C [17] in a modified rapid Golgi–Cox solution that contained 5% potassium dichromate, 5% potassium chromate and 5% mercuric chloride (Sigma-Aldrich) which was made 3 days prior to sacrifice as described previously [18]. Following incubation, 300 μm coronal sections were cut using a vibrating microtome (Hyrax V50, Carl Zeiss, Thornwood, NY, USA). Slices were then placed sequentially in 24-well plates filled with 30% (*w*/*v*) sucrose in 0.1 M phosphate buffered saline and processed and mounted in DPX medium (Sigma-Aldrich) as outlined previously [16,17,19]. 

### 2.3. Neurobiotin Electroporation

Rats (*n* = 6) were euthanized by sodium pentobarbitone overdose, decapitated and then the brain was quickly removed. Coronal sections (300 μm thick) containing the BLA were cut using a vibratome (Leica VT 1200S, Leica Biosystems, Newcastle, UK) in ice-cold high-Mg^2+^ Ringer solution, that contained (in mM): 130 NaCl, 3 KCl, 26 NaHCO_3_, 1.25 NaH_2_PO_4_, 5 MgCl_2_, 1 CaCl_2_, and 10 d-glucose [20]. Cut slices were transferred from ice-cold high-Mg^2+^ Ringer solution and incubated for 60 min in the same solution warmed to 34 °C in a water bath. The sections were then moved to normal Ringer solution (1 mM MgCl_2_, 2 mM CaCl_2_) and kept at room temperature (21–22 °C) for 30 min prior to start of labeling. All solutions were continually bubbled with carbogen gas. Patch electrodes were prepared as described previously [8] and filled with 2% neurobiotin™ (NB, Vector Labs, Burlingame, CA, USA) in an artificial intracellular solution containing (in mM): 135 Cs^+^MeSO_4_, 6 KCl, 1 EGTA, 2 MgCl_2_, 5 Na-HEPES, 3 ATP-Mg^2+^ and 0.3 GTP-Tris [21]. Neurons were electroporated, processed with Cy3-streptavidin (Sigma) and mounted in a glycerol based anti-fade media similarly to previous studies [8,22]. 

### 2.4. Neuronal Dendritic Morphology

All morphological properties of Golgi–Cox-impregnated and NB-filled cells were analyzed using Neurolucida™ software (MBF Bioscience Inc., Williston, VA, USA) in a manner similar to previous studies [8,22,23]. Golgi–Cox BLA principal neurons from a narrow bregma range between 2.5–3.2 mm [8,24] were traced in three-dimensions on a Zeiss Axioskop II (Carl Zeiss, Göttingen, Germany) with an automated *xyz* stage driven by Neurolucida, at 63X/1.4 NA magnification as for previous studies [16,17]. Confocal image mosaics of principal neurons filled with NB were acquired on a Leica TCS SP8 confocal microscope using a Leica NA 40X/1.3 NA air objective with a pixel size of 0.18 × 0.18 µm (*x* and *y* dimensions). Image stacks of between 100 and 180 images were acquired at a *z*-separation of 0.7 µm.

### 2.5. Statistical Methods

Mean and standard error of the mean (SEM) were calculated for each data set. For the purposes of this study, each individual principal neuron morphologically assessed is considered the *n,* from a total of 10 male Wistar rats. All statistical tests were calculated with Prism 7 (Graphpad, San Diego, CA, USA). When comparing the mean of two groups, if data were normally distributed (using a D’Agostino and Pearson normality test) and had no difference in their variances (using an *F* test), unpaired two tailed *t*-tests were used; otherwise, a non-parametric Mann–Whitney test was used, as indicated in the text. For branch order analysis, two-way ANOVAs, with Bonferroni post-tests were used, with method and branch order being the factors. Statistical significance was accepted at *p* < 0.05. All data are presented as mean ± SEM. 

## 3. Results

### 3.1. Neurobiotin-Filled Principal Neurons Have Longer Dendritic Arbor Length Compared to Golgi–Cox-Stained Principal Neurons within the BLA

We performed morphological analysis of NB-filled (*n* = 10) and Golgi–Cox-stained (*n* = 10) principal neurons, sampled from the basolateral subdivision of the BLA (Figure 1) in a manner identical to previous reports [8]. Principal cells were only included for analysis if their morphological properties were consistent with previous studies [8,25,26,27].

The total dendritic length (the summed length of the total neuronal arbor) was 93% larger for NB-filled, compared with Golgi–Cox-stained, BLA principal neurons (*p* < 0.0001 ****; Table 1, Figure 1). Total basal and apical dendrite length was 83% and 107% larger respectively, in BLA principal cells filled with NB, compared with those stained using Golgi–Cox (basal, *p* = 0.001 *; apical, *p* = 0.007 **; Table 1, Figure 1). We observed a ~3-fold increase in the number of dendritic nodes (bifurcations) and endings in NB-filled BLA principal neurons, compared with Golgi–Cox-stained neurons (*p* < 0.0001 ****; Table 1, Figure 1). Importantly, the mean tree length for basal dendrites (the mean length of each dendritic tree per cell, Figure 1) and the maximum apical terminal reach (measured distance from the soma to the longest apical dendritic termination) were not different between BLA principal neurons visualized using either method (Table 1).

### 3.2. Comparison of Spines Densities from NB-Filled and Golgi–Cox-Stained BLA Principal Neurons

The total spine density per 100 µm was not different between BLA principal neurons analysed using either method (Table 2). We also found no difference in the total basal and apical dendrite spine densities, when comparing the two methods (Table 2). In NB-filled neurons, the proximal (first- and second-order dendrites) spine densities of apical dendrites were 55% lower, while distal (third-order and later dendrites) spine densities were 40% higher, compared to Golgi–Cox-stained neurons (proximal, *p* = 0.0011 **; distal, *p* = 0.002 **; Table 2, Figure 2). Basal dendrite spine density did not significantly differ between NB-filled and Golgi–Cox-stained neurons (Table 2, Figure 2).

### 3.3. Branch Order Analysis of Dendritic Branch Segments, Dendritic Segment Length and Dendritic Spines from NB-Filled and Golgi–Cox-Stained BLA Principal Neurons

Following our analysis of the gross morphology of BLA principal neurons, we then investigated the branch order characteristics of basal and apical dendrites, by quantification of the number of dendritic segments per branch order, the mean length of dendritic segments per branch order, and the total number of spines per branch order of each dendritic tree type (Table 3).

The mean dendritic segment number per branch order was significantly higher in basal and apical dendrites of NB-filled BLA principal neurons, compared to Golgi–Cox-stained neurons (basal, *p* < 0.0001 ****; apical, *p* < 0.0001 ****; Table 3). Post-tests showed that this difference was significant for all first to fifth and beyond branch orders for basal dendrites and for all second to fifth and beyond branch orders for apical dendrites (Table 3). The mean length of dendritic segments per branch order was unchanged between NB-filled basal and apical dendrites compared to Golgi–Cox-stained BLA principal neuron dendrites (Table 3). Summary dendrogrammes of the branch number and length of individual segments clearly show that increased segment numbers at both apical and basal dendritic trees contribute to the overall difference in arbor lengths (Figure 3).

The mean dendritic spine density per branch order was not different for basal dendrites, but was significantly higher for apical dendrites of Golgi–Cox-stained compared to NB-filled BLA principal neurons (*p* = 0.0001 ****; Table 3). Post-tests showed that the mean apical dendritic spine densities of first- and second-order branches were higher in Golgi–Cox-stained neurons, compared to NB-filled neurons (Table 3). Apical spine densities were not significantly different for third-order branches (Table 3). The spine density of fourth- and fifth or greater branch orders, was significantly higher in NB-filled neurons, compared to Golgi–Cox-stained neurons (Table 3).

### 3.4. Sholl Analysis of Basal and Apical Dendrites from NB-Filled and Golgi–Cox-Stained BLA Principal Neurons

We then analysed the branching patterns of basal and apical dendrites, using the Sholl technique [15]. Sholl analysis showed a significant difference between NB and Golgi methods in dendritic interactions of basal (**** *p* < 0.0001; Figure 4) and apical (**** *p* < 0.0001; Figure 4) dendritic arbors. For NB-filled BLA principal cells, post-tests revealed that basal dendrites had a greater number of branches between 20–140 μm from the soma, compared to Golgi–Cox-stained neurons (Figure 4A). Similarly, the apical dendrites of NB-filled BLA principal cells had a greater number of branches between 80–240 μm from the soma, compared to Golgi–Cox-stained neurons (Figure 4B).

## 4. Discussion

Analysis of neuronal dendritic structure can provide insight into neuronal inputs and function. Varieties of methods have been used to delineate dendritic structures, but it remains unclear how the methods used may contribute to variability in structural quantification. We have addressed this issue by quantifying the morphological properties of BLA principal neurons filled with NB or stained using a modified Golgi–Cox method. We chose to compare quantitative data using both methods, based on previous studies reporting marked quantitative discrepancies for this neuronal type from rats of similar ages. We show that BLA principal neurons filled with NB have significant quantitative differences, compared to Golgi–Cox-stained neurons, including increased apical and basal dendritic length and branching complexity. In addition, we observed marked differences in apical spine density estimates between the two methods.

It is important to note that, even within our current study, the variation in tissue processing and mounting between the two techniques may contribute in part to the observed differences. In particular, Golgi–Cox techniques dehydrate the sample before the sectioning process. Thus, any tissue shrinkage that occurs during processing is consistent in *x-*, *y-* and *z-*planes. By contrast, with NB-filling, the sectioning takes place before the cell is labelled, and the tissue does not undergo dehydration. During mounting, Golgi–Cox sections are mounted in a solidifying medium, with shrinkage in the *z-*plane highly consistent [28] and minimal stretching/distortion in *x-* and *y-*planes. By contrast, NB-filled techniques require the slice to be mounted in an aqueous mounting media, with higher variability in the *z*-plane [13,14] and greater potential for stretching/distortion in *x*- and *y*-planes.

In recent morphological investigations of BLA principal neurons, total dendritic arbor lengths range from a low of 1338 µm [11] to a high of 7908 μm [14] in rats of similar ages (8.5–10-week-old Sprague-Dawley rats). Interestingly, large discrepancies in mean values have been consistently observed in studies that have implemented Golgi–Cox staining or dye-filling methods to quantify BLA principal cell morphology. In line with this, we found that the total dendritic arbor of NB-filled BLA principal neurons was ~2 fold larger than Golgi–Cox-stained neuronal arbors. Our branch order data demonstrated that this was primarily because NB-filling revealed significantly more dendritic branches across second to fifth and greater branch orders, compared to Golgi–Cox staining; this finding was consistent with increased radial Sholl interactions in NB-filled principal neurons. This demonstrates that the method used to delineate dendritic structures significantly influences quantitative measurements of dendrites.

In addition to these methodological differences in BLA principal neuron dendritic morphology, there are also significant disparities between studies using similar dye-filling techniques. Imaging fidelity remains the major limitation in fluorescent analysis, with the finer, more filamentous, dendritic processes captured at higher magnification. Using similar imaging acquisition methods, Ryan et al., 2016 reported a total dendrite length of 7908 μm for BLA principal cells from 8.5-week-old Sprague-Dawley rats filled with biocytin. Additionally, Faber et al., 2001 reported estimates for total dendrite length of 6299 to 6722 μm for principal neurons in the lateral amygdala from 17 to 20-day-old Wistar rats. A combination of species differences and animal age may, in part, contribute to the difference of these estimates, compared with our estimates, and those reported by others. Importantly, these two studies also applied correction factors to account for post-processing tissue shrinkage. Because we did not apply tissue correction factors to our samples, it is to be expected that our total dendrite estimates will be lower, compared to these previous reports. Further investigations are needed to determine the effect of post-processing and the accuracy of correction factors, which vary widely between individual samples [13,14], to account for variability in quantitative dendritic morphology resulting from tissue shrinkage. Comparing our current study to expected results from past studies in the BLA from our lab using Golgi–Cox [24] and NB-filling [8], indicates that the Golgi–Cox method is less variable than the NB method. Golgi–Cox-stained neurons in this study had ~10% greater total arbor length, ~10% greater basal arbor length and ~15% greater apical arbor length compared with our previous report [24]. With NB-filling, we observed a two-fold increase in dendrite lengths compared to our past results [8]. In the current study, the ability to discriminate signal from remarkably small voxel size, in this case, 0.26 µm in *x* and *y*, compared to 0.62 µm in our previous report [8], may underlie this difference. 

As mentioned previously, the distortion of the brain slice in aqueous mounting media is known to be highly variable [13], while less tissue shrinkage is observed with Golgi-impregnation [29,30,31]. This might, in part, contribute to the high variation in reports of BLA principal neuron arbor lengths using NB/dye-filling methods by us and others. Of note, there was low variability in the measurements of individual basal tree lengths and maximum terminal reach of apical dendrites between both techniques. Indeed, the ability of dye-filling methods that resolve more filamentous parts of the dendritic tree may aid in the classification of pyramidal cells within different brain regions [8], including the BLA [13]. Additionally, it is important to note that these differences may also be predicated on the ability of fluorescent confocal imaging to resolve the NB-filled material with less opacity than brightfield imaging.

We also compared spine densities of NB-filled and Golgi–Cox-stained principal neurons in the BLA. No overall difference in total spine density was found in BLA principal cells from both methods. A more nuanced approach revealed significantly higher spine densities in the apical proximal dendrites of Golgi–Cox-stained neurons compared to NB-filled cells, consistent with branch order analysis findings. A possible explanation for this difference may result from photon detector saturation of brighter proximal processes obliterating the fine spine structures close to the soma and thicker dendrites, which occurs when imaging is optimized for the thinner, less vivid smaller dendritic processes. We also found that the combined distal spine densities (third-order and greater) of NB-filled BLA principal cell apical dendrites were higher, compared to Golgi–Cox-stained cells. This reduction combined with reduced dendritic complexity in high order branch ramifications suggests that NB-filling provides a greater labeling efficiency of distal branch orders of BLA principal neurons compared to Golgi–Cox impregnation. This result might contribute to the consistent reporting of larger dendritic arbors of BLA principal neurons from dye-filling methods compared to Golgi–Cox studies [8,10,11,13,14,24,32].

In conclusion, our study has provided a detailed quantitative comparison of BLA principal neuron morphology using NB-filling and modified Golgi–Cox staining. Our morphological analyses have highlighted several quantitative differences with respect to general morphology and branch order characteristics of NB-filled and Golgi–Cox-stained BLA principal cells. Taken together, our findings indicate that NB-labeling provides a higher recovery of filamentous basal trees and the more distal accessory apical branches in BLA principal cells, leading to increased total dendrite arbor lengths and greater dendritic complexity, compared to that seen for Golgi–Cox impregnation. The increased efficacy of dye-filling compared to Golgi–Cox impregnation in distal branches is mirrored in the increased distal spine density of neurons assessed with NB. Importantly, the mean basal tree length and major central projections of the apical tree are preserved using both techniques, suggesting that mean basal tree length and maximal apical terminal lengths are particularly robust measures of BLA principal cell morphology. These results provide insights into differences between NB-filling and Golgi–Cox staining, and highlight methodological considerations contributing to quantitative and regional differences in BLA principal cell morphology reported in the literature. Furthermore, these results, combined with recent advancements in fluorescent microscopy, demonstrate a continued evolution and improvement in identifying and quantifying finer dendritic processes, allowing a more complete quantitative analysis of neuronal morphology.

## Figures and Tables

**Figure 1 brainsci-07-00165-f001:**
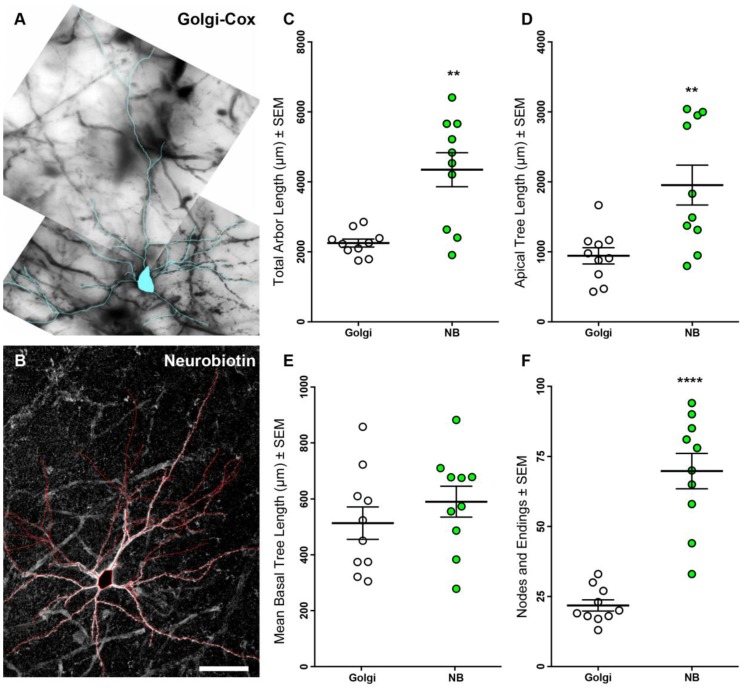
Basolateral amygdala (BLA) principal neuron dendritic arbors from Golgi–Cox and neurobiotin (NB)-filled methods. (**A**) shows a Golgi–Cox impregnated BLA principal neuron with the arbor identified using a magenta overlay; (**B**) shows a NB-filled BLA principal neuron with arbor identified using a red overlay; (**C**–**F**) show scatterplots of increased total dendritic arbor length, increased apical arbor length, unchanged mean basal tree length and increased dendritic nodes and endings in NB-filled (green circles) neurons compared to Golgi–Cox impregnated neurons (open circles). Nonparametric Mann–Whitney tests. ** *p* < 0.01 and **** *p* < 0.0001. Scale Bar: (**B**), 100 μm.

**Figure 2 brainsci-07-00165-f002:**
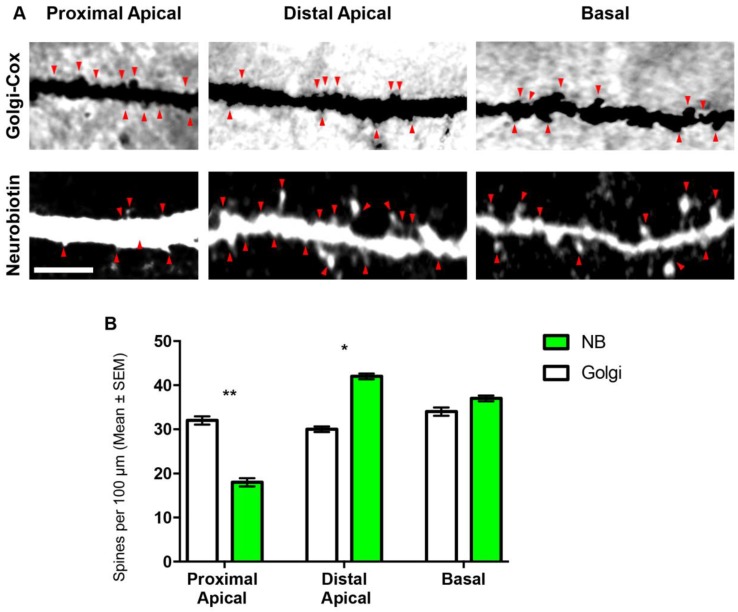
BLA principal neuron dendritic spine density from Golgi–Cox and NB-filled methods. (**A**) The top row shows representative Golgi–Cox, and the bottom row depicts representative NB-filled BLA principal neuron dendritic segments and dendritic spines (red arrowheads) from proximal apical, distal apical and basal dendrites; (**B**) shows a histogram comparing proximal apical, distal apical and total basal dendritic spine densities in Golgi–Cox-stained (open bars) and NB-filled (green bars) BLA principal cells. Unpaired Student’s two tailed *t*-tests, * *p* < 0.05, ** *p* < 0.01. Scale Bar: (**B**), 5 μm.

**Figure 3 brainsci-07-00165-f003:**
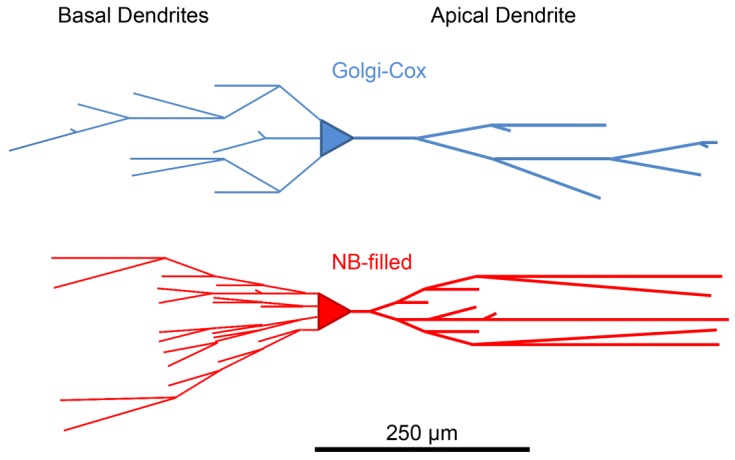
BLA principal neuron dendrogrammes of branch order segment number and length. Dendrogrammes show the mean number of apical and basal dendrites per branch order and the mean segment length per branch order in Golgi–Cox (blue) and NB-filled (red) BLA principal neurons. Note that the maximum apical terminal distance is unchanged between the two techniques. Increased overall length is likely due to increased efficacy of NB-filling of the more filamentous ramifications of the dendritic tree.

**Figure 4 brainsci-07-00165-f004:**
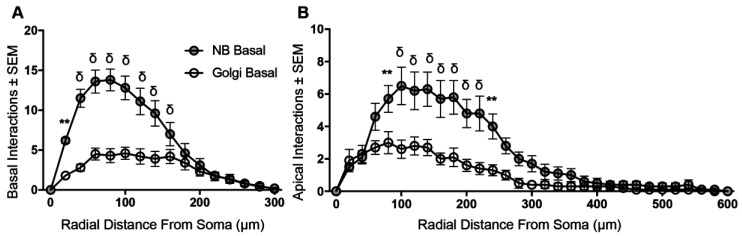
Sholl analysis shows increased dendritic branching in BLA principal dendritic arbors from NB-filled neurons compared to Golgi–Cox-stained neurons. (**A**,**B**) show Sholl analysis of increased dendritic branching in NB-filled (grey circles) neurons compared to Golgi–Cox impregnated neurons (open circles) in basal (method: **** *p* < 0.0001) and apical (method: **** *p* < 0.0001) arbors respectively. Two-way ANOVAs with Bonferroni post-tests, ** *p* < 0.01, and δ **** *p* < 0.0001.

**Table 1 brainsci-07-00165-t001:** General Morphology.

Parameter	Golgi–Cox (*n* = 10)	Neurobiotin (*n* = 10)	*p*-Value
Total dendritic length (µm)	2251 ± 113	4349 ± 488	0.003 ** ^a^
Apical dendritic length (µm)	944 ± 116	1955 ± 285	0.007 ** ^a^
Basal dendritic length (µm)	1307 ± 102	2394 ± 278	0.01 * ^a^
Mean basal tree length (µm)	512 ± 58	589 ± 55	0.35
Nodes and endings	22 ± 2	70 ± 6	<0.0001 **** ^a^
Maximum apical terminal length (µm)	484 ± 51	431 ± 51	0.47

Unpaired Student’s two tailed *t*-test or ^a^ Mann–Whitney nonparametric tests, * *p* < 0.05, ** *p* < 0.01 and **** *p* < 0.0001.

**Table 2 brainsci-07-00165-t002:** Dendritic Spine density (spines per 100 μm).

Parameter	Golgi–Cox (*n* = 10)	Neurobiotin (*n* = 10)	*p*-Value
Total spine density	33 ± 2	38 ± 2	0.18
Apical spine density	31 ± 2	38 ± 2	0.16
Basal spine density	34 ± 3	37 ± 2	0.79
Proximal basal spine density	37 ± 4	30 ± 3	0.09
Distal basal spine density	32 ± 3	38 ± 3	0.12
Proximal apical spine density	40 ± 5	18 ± 3	0.0011 **
Distal apical spine density	30 ± 2	42 ± 2	0.002 *

Unpaired Student’s two tailed *t*-test, * *p* < 0.05, ** *p* < 0.01.

**Table 3 brainsci-07-00165-t003:** Branch order characteristics for Golgi–Cox-stained and neurobiotin-filled BLA principal cells. All data presented as mean ± SEM.

Branch Order Properties	Golgi–Cox Basal	Neurobiotin Basal	Golgi–Cox Apical	Neurobiotin Apical	Adjusted *p-*Values
1st order branch segments	2.7 ± 0.1	4.0 ± 0.1	1.0 ± 0.0	1.0 ± 0.0	B: *p* = 0.001 *** A: *p* > 0.99
1st order mean branch segment length (μm)	64 ± 20	19 ± 4	72 ± 21	21 ± 7	B: NS A: NS
1st order branch mean spine density	44.5 ± 5.7	17.4 ± 3.0	47.1 ± 1.8	11.4 ± 1.1	B: NS A: *p* < 0.0001 ****
2nd order branch segments	4.6 ± 0.2	7.8 ± 0.1	2.0 ± 0.0	2.0 ± 0.0	B: *p* < 0.0001 **** A: *p* > 0.99
2nd order mean branch segment length (μm)	76 ± 12	49 ± 6	91 ± 15	31 ± 7	B: NS A: NS
2nd order branch mean spine density	34.1 ± 2.9	32.0 ± 2.7	38.5 ± 1.6	21.8 ± 1.2	B: NS A: *p* < 0.0001 ****
3rd order branch segments	4.2 ± 0.2	11.1 ± 0.4	2.9 ± 0.1	3.5 ± 0.1	B: *p* < 0.0001 **** A: *p* < 0.004 **
3rd order mean branch segment length (μm)	109 ± 14	56 ± 3	134 ± 37	37 ± 5	B: NS A: NS
3rd order branch mean spine density	32.1 ± 3.0	36.2 ± 2.9	31.1 ± 0.7	30.9 ± 1.5	B: NS A: *p* > 0.99
4th order branch segments	1.8 ± 0.2	9.8 ± 0.5	2.6 ± 0.1	6.1 ± 0.2	B: *p* < 0.0001 **** A: *p* < 0.0001 ****
4th order mean branch segment length (μm)	64 ± 29	65 ± 5	108 ± 44	64 ± 8	B: NS A: NS
4th order branch mean spine density	32.9 ± 6.1	40.7 ± 2.6	29.1 ± 1.2	38.4 ± 1.5	B: NS A: *p* < 0.0001 ****
≥5th order branch segments	1.0 ± 0.1	4.0 ± 0.2	2 ± 0.1	4.5 ± 0.2	B: *p* < 0.0001 **** A: *p* > 0.99
≥5th order mean branch segment length (μm)	79 ± 38	134 ± 27	19 ± 12	286 ± 46	B: NS A: *p* < 0.0001 ****
≥5th order mean spine density	33.6 ± 3.2	45.3 ± 4.4	34.1 ± 1.9	40.2 ± 1.1	B: NS A: *p* = 0.01 *

Golgi principal neuron *n* = 10, NB principal neuron *n* = 10. * *p* < 0.05, ** *p* < 0.01, *** *p* < 0.001, **** *p* < 0.0001, two-way ANOVAs with Bonferroni’s post-test (the adjusted *p*-value reported in the table for parameters that had significant method effects) for branch order and experimental method for branch segment number (Basal: Branch order: **** *p* < 0.0001, Method: **** *p* < 0.0001; Apical: Branch order: **** *p* < 0.0001, Method: **** *p* < 0.0001), mean branch segment length (Basal: Branch order: * *p* = 0.02, Method: *p* = 0.27; Apical: Branch order: *** *p* = 0.001, Method: *p* < 0.85) and branch segment spine density (Basal: Branch order: **** *p* < 0.0001, Method: *p* = 0.17; Apical: Branch order: **** *p* < 0.0001, Method: **** *p* < 0.0001). Abbreviations: B: Basal; A: Apical; NS: Not Significant.

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
