# Peer review of "Investigating Methodological Differences in the Assessment of Dendritic Morphology of Basolateral Amygdala Principal Neurons—A Comparison of Golgi–Cox and Neurobiotin Electroporation Techniques"

_brainsci, 2017, doi:10.3390/brainsci7120165_

Round 1

Reviewer 1 Report

In this sudy, different staining methods were applied in order to visualize the basolateral amyglada (BLA) principal neuron morphology quantitatively. Together with the specifications of above-mentioned techniques, modified Golgi-Cox staining and neurobiotin (NB) staining, the results revealed with this study was compared well with the existing literature. Methods utilized for the staining processed with modifications were given in detail and written clearly. Moreover the experiments conducted seem to be designed according to properties of BLA principle neurons. In general, NB staining was concluded to give much more distinguished differences compared to other staining, which were explained in detail in discussion part. In terms of brach order characteristics, NB-labelling seemed to result in greater dendritic complexity, which later on is expected to enable much more quantitative analysis for the nuronal morphology. 

 The manuscript is well-written and the experimental part is given in detail. Followings are some comments about this study:   

 1. In introduction part, maybe it would be better to mention about the age of animals used in reference 8, so that the given result is more comparable with the following results in the next sentence (third paragraph in intro part). 

2. In Figure 1, standard deviations are more for data points of NB-labelling compared to Golgi-Cox staining. What would be authors’ comments for that? 

3. Results are given clearly and compared within each other well. For regional quantification, it would be much clearer once it was highlighted if obtained results for both apical and basal regions were normalized with respect to average expected numbers, if there is any?

 4. More confocal images (for neuronal arbors) might be more helpful to understand the comparision of two techniques better in terms of brach orders.

 5. Apart from improved image processing capabilities, do authors think that different (step by step) procedures for the staining (revealed in experimental section) might be contributed to better visualization of neuronal dendrities?

Author Response

We thank the reviewers for their thoughtful comments. We are delighted to submit a revised manuscript, including two new figures and hope that our revision provides an adequate clarification of the present work and makes it suitable for publication.

Reviewer 1

In this sudy, different staining methods were applied in order to visualize the basolateral amyglada (BLA) principal neuron morphology quantitatively. Together with the specifications of above-mentioned techniques, modified Golgi-Cox staining and neurobiotin (NB) staining, the results revealed with this study was compared well with the existing literature. Methods utilized for the staining processed with modifications were given in detail and written clearly. Moreover the experiments conducted seem to be designed according to properties of BLA principle neurons. In general, NB staining was concluded to give much more distinguished differences compared to other staining, which were explained in detail in discussion part. In terms of brach order characteristics, NB-labelling seemed to result in greater dendritic complexity, which later on is expected to enable much more quantitative analysis for the nuronal morphology. 

 The manuscript is well-written and the experimental part is given in detail. Followings are some comments about this study:   

1.     In introduction part, maybe it would be better to mention about the age of animals used in reference 8, so that the given result is more comparable with the following results in the next sentence (third paragraph in intro part).

We have identified the age of the rats studied in [8] so as to provide context for the comparisons that follow.

This information is provided in the manuscript in line 84.

2.     In Figure 1, standard deviations are more for data points of NB-labelling compared to Golgi-Cox staining. What would be authors’ comments for that?

NB-filling has consistently exhibited larger dendritic arbors compared with Golgi-Cox staining. As mentioned in the introduction, Golgi-Cox staining has revealed total dendritic arbors lengths of BLA principal neurons from 8-10 week old rats in the range of 1300 – 1800 µm [1, 2]. A recent Golgi-Cox study from our lab reported a total dendrite length of 1928 µm [3], which is in line with this study and previous reports using similarly aged-rats. In studies using dye-filling methods we have previously reported a BLA principal neuron dendrite length of 2034 µm [4] and 4349 µm in this study, while others have noted lengths of up to 7908 µm in rats of similar ages [5]. These results highlight significantly greater variations in the total dendrite length of BLA principal cells reported from dye-filling studies compared to those reported from Golgi-Cox studies. This is likely due to increased penetration of NB into the more filamentous dendrites. This is based on our observation of the maximum terminal reach of the major trunks of apical dendrites which show no difference in either mean or variability between methods. Similarly, the mean basal dendritic tree length and their variability are not different between techniques. Thus, the variability is likely due to more efficient labeling of filamentous accessory branches in NB-filled neurons compared to Golgi-Cox stained neurons. Other methodology considerations that might contribute to these differences, such as reduced variation in shrinkage of Golgi-Cox stained tissue, and differences in imaging, as evidenced by changes in voxel sizes between techniques are also elaborated upon in the discussion.

This information is provided in our manuscript in lines 271-281, 316-336 and 360-369.

Figure 1 now reflects mean basal tree lengths to be more informative of the above point.

3.     Results are given clearly and compared within each other well. For regional quantification, it would be much clearer once it was highlighted if obtained results for both apical and basal regions were normalized with respect to average expected numbers, if there is any?

Comparisons of our data, compared to expected results from within our own lab have been elaborated upon. We report relative changes from these works and our current study to help provide further context to our study.

This information has been provided in the discussion lines 316-324

4.     More confocal images (for neuronal arbors) might be more helpful to understand the comparision of two techniques better in terms of brach orders.

We appreciate the reviewer’s comments and agree that a visual representation of segmental data will improve our manuscript. A dendrogramme – visualizing the mean segment number and length at each branch order for both apical and basal dendrites has been included in our revised submission.

This addition appears as new figure 3 and its associated figure legend lines 522-528. Further reference is provided in the results lines 223-226 and summarized in discussion lines 360-369.

5.     Apart from improved image processing capabilities, do authors think that different (step by step) procedures for the staining (revealed in experimental section) might be contributed to better visualization of neuronal dendrities?

We thank the reviewer for bringing this excellent point to our attention. We have considered the methodological differences that may contribute to different visualization results under two broad headings; tissue processing and mounting procedures. In particular, aqueous mounting of dye-filled material has a higher degree of variability in z-plane distortion/shrinkage, as evidenced by the large range of correction factors used [5, 6] compared to Golgi-Cox staining [7-10].  

This information has been included in the methods, lines 121 and lines 140, and in the discussion, lines 270-281 and 340-353.

Reviewer 2 Report

In this study, Klenowski and colleagues used two methods to investigate the dendritic morphology of BLA and did a series of analyses to reveal the differences between them. Overall, this paper was well written with sufficient introduction and discussion. However, I still have some concerns about this work.

1.According to Line 93-100, the authors used two different microscopes with different magnification to capture images from Golgi-Cox and NB method. Just as they mentioned in the discussion, those differences could lead to discrepant results. It's possible that all the differences that they found in this study are just the differences between two image acquisition systems. The authors should explain and discuss this point in the paper.

2. The "n" of statistics is not very clear. The authors only mentioned a total number of 10 male rats were used in this study (Line102-104). They should mention how many neurons from each animal were analyzed and how many animals were used respectively for Golgi-Cox and NB method.

3. Figure 1C-F: It seems that the NB method had a greater variation that Golgi-Cox. Could the authors explain and discuss this point?

4. Table 2: Could the authors also provide the images of dendritic spines?

Author Response

We thank the reviewers for their thoughtful comments. We are delighted to submit a revised manuscript, including two new figures and hope that our revision provides an adequate clarification of the present work and makes it suitable for publication. 

Reviewer 2

In this study, Klenowski and colleagues used two methods to investigate the dendritic morphology of BLA and did a series of analyses to reveal the differences between them. Overall, this paper was well written with sufficient introduction and discussion. However, I still have some concerns about this work.

1.According to Line 93-100, the authors used two different microscopes with different magnification to capture images from Golgi-Cox and NB method. Just as they mentioned in the discussion, those differences could lead to discrepant results. It's possible that all the differences that they found in this study are just the differences between two image acquisition systems. The authors should explain and discuss this point in the paper.

We acknowledge this point outlined, quite rightly, by the reviewer and have outlined the difference in the imaging in the discussion. To illustrate that NB-filling provides more elaborate filling of filamentous accessory dendrites, we have interpreted our data in light of stable unchanged maximal apical terminal distances between the two techniques, and stable, unchanged mean basal tree lengths between the two techniques. I.e. the thicker main apical truck and the thicker basal arbors are not different between both methodologies, thus, based on our data, our interpretation is that the extra length of NB-filled dendrites is provided by the filamentous distal branches being more readily filled and imaged using NB (Also see response to Reviewer 1, comment 1). A representative dendrogramme of Golgi-Cox and NB-filled BLA neurons provides a visualization of the data provided in table 3, in order to clarify this point. We note that improved imaging between fluorescent and brightfield imaging may also contribute to our results.

This information appears as new figure 3 in our resubmission, with the figure legend in the manuscript lines 522-528.

Additional results in lines 223-226 and discussion points are in lines 316-336 and 360-369.

2. The "n" of statistics is not very clear. The authors only mentioned a total number of 10 male rats were used in this study (Line102-104). They should mention how many neurons from each animal were analyzed and how many animals were used respectively for Golgi-Cox and NB method.

We used 4 rats for Golgi-cox assessments and 6 rats for NB-filling. Of these, n=10 neurons from each technique were assessed, from within a narrow bregma of 2.7 – 3.2 mm, in line with past studies [4]. A minimum of 2 neurons per animal were traced.

This information is included in the methods lines 112, 124 and 147.

3. Figure 1C-F: It seems that the NB method had a greater variation that Golgi-Cox. Could the authors explain and discuss this point?

We thank the reviewer for bringing up this point. As mentioned in the response to reviewer 1, comment 1, our studies and work form other labs have shown that dye-filling methods consistently report larger overall dendritic arbors and provide significantly larger variations in total dendrite arbor length of BLA principal neurons compared to those reported from Golgi-Cox studies. Our revised manuscript now provides additional discussion points on methodological differences that may result in different visualization results under two broad headings; tissue processing and mounting procedures. In particular, aqueous mounting of dye-filled material has a higher degree of variability in z-plane distortion/shrinkage, as evidenced by the large range of correction factors used [5, 6] compared to Golgi-Cox staining [7-10].  Differences in imaging, as evidenced by changes in voxel sizes between techniques are also elaborated upon in the discussion.

This information has been included in the methods, lines 121 and lines 140, and in the discussion, lines 270-281, 325-333, 350-353 and 360-369.

4. Table 2: Could the authors also provide the images of dendritic spines?

We have provided a new figure including representative images of apical proximal, apical basal and basal dendritic spines from NB-filled and Golgi-Cox stained BLA principal neurons.

This has been included as new figure 2 in our revised submission, and its figure legend in the manuscript, lines 514-520.

Round 2

Reviewer 2 Report

Overall, I am satisfied with the revised version. I only have two minor points:

1. Figure 2A: The authors should use arrows to indicate the spines in the images.

2. Table 3: I am a little confused with some data. For example, in the "≥5th order mean spine density", there is no significant difference in Basal and P=0.01 for Apical. But the mean values ofBasal are different while the mean values of Apical are identical. Could the authors double check the data and statistics of Table 3?

Author Response

1. Figure 2A: The authors should use arrows to indicate the spines in the images.

Revision #2 - we have added red arrowheads for spine identification in figure 2.

2. Table 3: I am a little confused with some data. For example, in the "≥5th order mean spine density", there is no significant difference in Basal and P=0.01 for Apical. But the mean values ofBasal are different while the mean values of Apical are identical. Could the authors double check the data and statistics of Table 3?"

Revision #2 - we have corrected the >5th order branch mean spine densities in the branch order table and it is now reported correctly with SEMs instead of SD.  The >5th order branch segment numbers also contained an error; this has been corrected, with SEM.